# Anticancer Effect of E26 Transformation-Specific Homologous Factor through the Induction of Senescence and the Inhibition of Epithelial–Mesenchymal Transition in Triple-Negative Breast Cancer Cells

**DOI:** 10.3390/cancers15215270

**Published:** 2023-11-02

**Authors:** Soyoung Lim, Jihyun Lim, Aram Lee, Keun-Il Kim, Jong-Seok Lim

**Affiliations:** Department of Biological Sciences, Research Institute of Women’s Health, Sookmyung Women’s University, Seoul 04310, Republic of Korea; sornate@sookmyung.ac.kr (S.L.); jhlim@sookmyung.ac.kr (J.L.); 1215728@sookmyung.ac.kr (A.L.); kikim@sookmyung.ac.kr (K.-I.K.)

**Keywords:** EHF, NDRG2, STAT3, senescence, breast cancer, TCGA data

## Abstract

**Simple Summary:**

N-myc downstream-regulated gene 2 (NDRG2) is a tumor suppressor in breast cancer, and its expression is positively correlated with ETS homologous factor (EHF) expression. To determine the functional role of EHF in breast cancer cells, EHF was overexpressed in breast cancer cells, which resulted in growth retardation and an increased apoptotic rate by chemotherapeutic agent treatment. Moreover, cellular senescence was induced, and the migration ability was suppressed by EHF overexpression. Signal transducer and activator of transcription 3 (STAT3) signaling was also suppressed by EHF overexpression. The subcutaneous injection of 4T1-EHF cells into Balb/c mice resulted in smaller tumor sizes compared to 4T1-mock tumors, and pulmonary metastasis was also suppressed by EHF overexpression. Patients with a high EHF expression showed a better prognosis than patients with low EHF expression. These results suggest that EHF might be a tumor suppressor gene in breast cancer cells and might be a prognostic marker of breast cancer.

**Abstract:**

The aim of the present study was to evaluate the effect of ETS homologous factor (EHF) in malignant breast cancer cells. The overexpression and knockdown of the EHF gene in human and mouse breast cancer cells were performed, and the TCGA dataset and Q-omics were analyzed. We found that the tumor suppressor NDRG2 is correlated with EHF gene expression in triple-negative breast cancer cells, that EHF overexpression results in reduced cell proliferation and that apoptosis is promoted by the chemotherapeutic reagent treatment of EHF-overexpressing cells. By EHF overexpression, senescence-associated β-galactosidase activity and p21^WAF1/CIP1^ expression were increased, suggesting that EHF may induce cellular senescence. In addition, the overexpression of EHF reduced the migratory ability and inhibited epithelial–mesenchymal transition (EMT). Furthermore, EHF inhibited the phosphorylation of STAT3. The overexpression of EHF also reduced the tumor size, and lung metastasis in vivo. At the tumor site, β-galactosidase activity was increased by EHF. Finally, the Kaplan–Meier-plotter analysis showed that TNBC patients with a high expression of EHF had a longer relapse-free survival rate. Our findings demonstrated that EHF inhibits breast tumor progression by inducing senescence and regulating EMT in TNBC cells.

## 1. Introduction

Cellular senescence is permanent cell cycle arrest induced by telomere shortening, DNA damage, epigenetic alterations, and various stressors [1]. Senescent cells are unable to reenter the cell cycle, and their proliferation ceases. Senescence occurs in a variety of situations, including injury, development, tissue remodeling, and cancer [2]. In cancer, senescence is generally considered to be a tumor-suppressive mechanism by inhibiting tumorigenesis and tumor progression. Senescence is maintained by the p16^INK4A^-pRB and/or p53-p21^WAF1/CIP1^ tumor pathway [3]. p16^INK4A^ acts upstream of pRB, and p21^WAF1/CIP1^ acts downstream of p53. They are negative regulators of the cell cycle and crucial cyclin-dependent kinase inhibitors (CDKIs). Triple-negative breast cancer (TNBC) is a subtype of breast cancer that is the leading cause of death in breast cancer patients worldwide. TNBC cells are characterized by the loss of estrogen receptor (ER), progesterone receptor (PR), and human epidermal growth factor receptor 2 (HER2) [4]. TNBC is a highly malignant breast cancer subtype with invasive and migratory properties. As a result, TNBC patients have a worse prognosis than other breast cancer subtypes. N-myc downstream-regulated gene 2 (NDRG2) belongs to the NDRG2 family, which consists of four members, NDRG1, NDRG2, NDRG3, and NDRG4. They are intracellular proteins that are involved in cell differentiation, proliferation, stress response, and development [5]. Among this family, NDRG2 is known as a tumor suppressor gene that is associated with tumor incidence, metastasis, and progression [6]. The expression of NDRG2 is downregulated in various cancers, including adult T-cell leukemia-lymphoma (ATLL), breast cancer, prostate cancer, lung cancer, and thyroid cancer [7,8,9,10,11]. Our previous study revealed that the overexpression of NDRG2 suppressed the migration and invasion of breast cancer cells by repressing the STAT3/Snail signaling pathway, which plays an important role in cell survival, mortality, and proliferation [12]. Based on these findings, we investigated the target genes of NDRG2. The analysis of the TCGA dataset revealed a positive correlation between NDRG2 and EHF in TNBC patients. ETS homologous factor (EHF), also known as epithelial-specific expression 3 (ESE3), is a member of the E26 transformation-specific (ETS) family. It binds to a conserved DNA site, known as the ETS binding site (EBS; GGAA/T), in the promoter and enhancer of target genes [13]. ESE subfamily members (ELF3, ELF5, EHF, and SPDEF) share an epithelial-specific profile and N-terminal pointed (PNT) domain, which is involved in transcriptional repression, oligomerization, and protein–protein interactions [14,15]. Recent studies have described its involvement in various cancers, as a tumor suppressor gene or an oncogene, depending on the tumor type. While previous studies have revealed the tumor-promoting role of EHF in thyroid and ovarian cancer [16,17], EHF inhibits the invasion and metastasis of pancreatic cancer cells by directly binding to the EBS in the promoter region of the E-cadherin gene and upregulating its expression. In addition, in prostate cancer, EHF binds to the EBS in the promoter region of IL-6 and represses its transcription, resulting in STAT3 inactivation [18]. However, the role of EHF in breast cancer and its relationship with NDRG2 remain unknown. Therefore, we investigated the anticancer effect of the EHF overexpression and further explored the underlying mechanisms of the EHF effects in TNBC cells.

## 2. Materials and Methods

### 2.1. Cell Culture and Reagents

The human breast cancer cell lines MDA-MB-231 and MDA-MB-453 were purchased from the American Type Culture Collection (ATCC, Manassas, VA, USA). The Hs578T and BT549 human breast cancer cell lines were kindly provided by Dr. Sukjoon Yoon (Sookmyung Women’s University, Seoul, Republic of Korea). 4T1 cells were kindly provided by Professor KD Kim (Gyeongsang National University, Jinju, Republic of Korea). Cells were maintained in Dulbecco’s modified Eagle’s medium (DMEM; Gibco/Invitrogen, Carlsbad, CA, USA) containing 10% fetal bovine serum (Gibco/Invitrogen), 100 U/mL penicillin, and 100 μg/mL streptomycin (Gibco/Invitrogen). All cells were grown in a humidified 5% CO_2_ incubator at 37 °C. Paclitaxel and doxorubicin were purchased from Sigma-Aldrich (St. Louis, MO, USA).

### 2.2. Mouse and Tumor Models

Female Balb/c mice (6 to 8 weeks old) were purchased from Daehan Bio link (Eumseong, Chungcheongbuk-Do, Republic of Korea). Tumor models were generated by the subcutaneous injection of 5 × 10^5^ 4T1 cells into the mammary fat pad or intravenous injection of 10^5^ 4T1 cells. Tumor volumes were measured using the formula (A × B^2^)/2 (mm^3^) (A > B). Tumor-bearing mice were humanely sacrificed within one month. For the ex vivo Western blot analysis using tumor tissue, the tumors were harvested and diced into pieces. Then, pieces of tumor were homogenized to acquire tumor cells. The mice were maintained in specific pathogen-free facilities at Sookmyung Women’s University. All animal experiments were performed in accordance with the guidelines of the Institutional Animal Care and Use Committee and were approved by the Institutional Ethics Committee of Sookmyung Women’s University.

### 2.3. Plasmid Transfection and Overexpression of EHF

For transient transfection, cells were transfected with pCMV-Taq-2B vector (Origene, Rockville, MD, USA) or pCMV6-Entry-EHF (Origene) using Lipofectamine™ 3000 (Invitrogen) according to the manufacturer’s instructions. For stable 4T1-EHF and MDA-MB-231-EHF clones, cells were transfected with pCMV6-Entry-EHF (Origene) using Lipofectamine™ 3000 (Invitrogen) and selected with 1 mg/mL G418 (Duchefa Biochemie, Haarlem, The Netherlands) in a complete medium.

### 2.4. Determination of Cell Growth

To analyze the growth of MDA-MB-231 and MDA-MB-453 cells, the cells were seeded at 70–80% confluence in medium without antibiotics for 12 h and transiently transfected with the pCMV6-Entry plasmid containing the EHF gene using Lipofectamine™ 3000 (Invitrogen). After incubation for 24 h, the cells were seeded in a 96-well plate (5 × 10^3^ cells per well) and incubated in IncuCyte ZOOM^®^ (Sartorius, Göttingen, Germany). The average phase object area (μm^2^) was calculated to analyze the relative cell confluence.

### 2.5. RNA Isolation and Quantitative Real-Time RT-PCR

RNA was extracted from harvested cells using TRIzol reagent (Molecular Research Center Inc, Cincinnati, OH, USA). Five micrograms of RNA were reverse transcribed into cDNA. Oligo-dT, RT buffer (Promega, Madison, WI, USA), dNTPs (Bioneer, Daejeon, Republic of Korea), and M-MLV reverse transcriptase (Promega) were used for cDNA synthesis. Quantitative real-time RT-PCR was performed using an ABI StepOnePlus™ real-time PCR thermal cycler and Power SYBR Green PCR Master Mix (Applied Biosystem, Foster City, CA, USA). Table 1 shows the sequences of the specific primers used in this study. Glyceraldehyde-3-phosphate dehydrogenase (GAPDH) and cyclophilin were used as endogenous controls. Triplicate experiments were performed for validation.

### 2.6. Knockdown of EHF Expression by Small Interfering RNA (siRNA)

EHF-specific siRNAs were purchased from Sigma-Aldrich. Control siRNA was purchased from Santa Cruz Biotechnology (Santa Cruz, CA, USA). The siRNAs purchased were transfected into MDA-MB-231 cells using Lipofectamine™ RNAiMAX (Invitrogen) according to the manufacturer’s instructions.

### 2.7. Western Blot

Cells were harvested and washed with Dulbecco’s phosphate-buffered saline (DPBS; WELGENE, Daegu, Republic of Korea). Proteins were isolated using Pro-Prep™ reagent (iNtRON Biotechnology, Seongnam, Republic of Korea). Proteins were quantified by the Bradford (BIO-RAD, Hercules, CA, USA) assay and mixed with 5× sample buffer. Total proteins were separated by electrophoresis on gradient SDS-PAGE gels (8% to 15%) and transferred onto a PVDF membrane (Amersham Biosciences, Bukres, UK). Membranes were incubated with Tris-buffered saline plus 0.05% Tween-20 (TBST) with 5% skim milk for blocking and were incubated with primary antibodies overnight at 4 °C. AMPK, GAPDH, NDRG2, Snail, Slug, E-cadherin, and α-actinin antibodies were purchased from Santa Cruz Biotechnology. α-Tubulin and EHF antibodies were purchased from Thermo Fisher Scientific (Waltham, MA, USA). Antibodies against N-cadherin, p21^WAF1/CIP1^, p-AMPK, STAT3, and p-STAT3 were purchased from Cell Signaling Technology (Beverly, MA, USA). The Flag antibody was purchased from Sigma-Aldrich. After incubation with primary antibodies, the membranes were washed and incubated with HRP-conjugated secondary antibodies. The blots were visualized by the enhanced chemiluminescence system using an Ez-Capture MG (ATTO Corporation, Tokyo, Japan).

### 2.8. Wound Healing Assay

Cells were seeded in 6-well plates and transiently transfected with pCMV-Taq-2B or pCMV6-Entry-EHF using Lipofectamine™ 3000. After incubation for 6 h, the cells were washed with DPBS, and incubated in starvation medium with mitomycin C (Sigma-Aldrich) for 4 h to rule out the effect of cell proliferation. The cell monolayer was scratched with a sterile 1000 μL pipette tip and washed with DPBS to remove detached cells, and four randomly selected sites were photographed.

### 2.9. Transwell Assay

Migration and invasion assays were performed using 24-well Transwell units with 6.5 mm diameter and 8.0 μm pore size polycarbonate filters (Corning Costar, Cambridge, MA, USA). For the invasion assay, the cells were starved overnight in serum-free medium with mitomycin C (Sigma-Aldrich) for 4 h to rule out the effect of cell proliferation. The Transwell was placed over DMEM with 10% FBS and coated with 20 μL of 1:2 mixture of Matrigel-DMEM (Matrigel; BD Biosciences, San Jose, CA, USA) as an intervening invasive barrier. A total 5 × 10^4^ cells were suspended in serum-free DMEM and added to the upper part of the Transwell. The cells were incubated for 24~48 h at 37 °C and the remaining cells in the upper part of the Transwell were removed using a cotton swab. The cells that migrated to the lower surface of the Transwell were stained with 0.1% crystal violet/2% ethanol (*w*/*v*) solution. Acetic acid (10%) (Sigma-Aldrich) was used to extract the blue dye and the migrated cells were measured using a SpectraMax i3x microplate reader (Molecular Devices, San Jose, CA, USA) by absorbance at 590 nm. For the migration assay, the described experiments were performed without Matrigel coating.

### 2.10. Senescence-Associated β-Gal Assay

MDA-MB-231 and 4T1 cells were seeded on the plate and stained with the senescence-associated β-galactosidase (SA β-gal) staining kit (Cell Signaling, #9860) according to the manufacturer’s instructions. Briefly, cells were fixed for 15 min at room temperature, washed with DPBS, and stained with 1× β-galactosidase detection solution for overnight at 37 °C. The next day, the cells were washed and overlaid with 70% glycerol. Images were captured using a bright light microscope (10× magnification objective) (IX71, OLYMPUS, Tokyo, Japan). For the tissue staining, tumor tissue sections were frozen and stained for SA β-gal. The next day, the sections were counterstained for nuclear fast red. Images were taken with a bright light microscope (40× magnification objective) (RVL2-K2, Echo, San Diego, CA, USA).

### 2.11. Annexin V/PI Apoptosis Assay

MDA-MB-231 and MDA-MB-453 cells were seeded in 6-well plates and transiently transfected with pCMV-Taq-2B or pCMV6-Entry-EHF using Lipofectamine™ 3000. Alternatively, stable MDA-MB-231-EHF and stable 4T1-EHF cells were seeded. After 24 h of incubation, the cells were treated with the indicated concentrations of paclitaxel or doxorubicin for 24 h or 48 h. The cells were then harvested and incubated with Annexin V and PI (Apoptosis Detection Kit, BD Biosciences) for 10 min. Apoptotic cells were evaluated by FACSCanto II flow cytometer (BD Biosciences). The flow cytometry data were analyzed by FlowJo software (Tree Star, Ashland, OR, USA).

### 2.12. Survival Analysis

The relapse-free and overall survival were plotted using the Kaplan–Meier plotter (http://kmplot.com/analysis/index.php?p=service) (accessed on 2 November 2020) [19]. To generate the survival plot of TNBC patients, breast cancer patients that are negative for an estrogen receptor, progesterone receptor, and HER2 were selected. The disease-free interval analysis was performed using the Xenabrowser (https://xenabrowser.net/) (accessed on 28 August 2023).

### 2.13. Dataset Analysis

The “BRCA.exp.547.med.txt.” file from TCGA breast cancer online (https://tcga-data.nci.nih.gov/docs/publications/brca_2012/) (accessed on 30 November 2020) was used for microarray gene expression data of NDRG2 and EHF. Q-omics software (version 1.5) was used to generate NDRG2 and EHF correlation graphs for TNBC and non-TNBC [20]. The NDRG2 and EHF correlation graph of breast cancer cell lines was generated using the UCSC Xena platform (https://xenabrowser.net/) (accessed 30 November 2022) [21]. The graph of the pathological T and N stage was generated using the TCGA dataset from the UCSC Xena platform. Heatmaps of NDRG2 and EHF were generated using the SEEK platform (https://seek.princeton.edu/seek/) (accessed 20 April 2022) [22]. Heatmaps of EHF, CDH1, SNAI1, and SNAI2 were generated using the SEEK platform and Q-omics.

### 2.14. Statistical Analysis

Statistical significance was analyzed by the student’s *t*-test and one-way ANOVA (Tukey’s post-test for multiple comparisons) using GraphPad prism software version 10 (GraphPad Software, San Diego, CA, USA). Values represent the mean ± SD. A value of *p* < 0.05 was considered significant. * *p* < 0.05, ** *p* < 0.01, *** *p* < 0.001, **** *p* < 0.0001.

## 3. Results

### 3.1. The Positive Correlation between NDRG2 and EHF Gene Expression Levels in Breast Cancer

In breast cancer cells, several studies have identified NDRG2 as a tumor suppressor gene, as it is involved in tumor cell proliferation, migration, invasion, and angiogenesis [23,24,25]. We previously reported that NDRG2 suppresses EMT in breast cancer cells [12] and sought to identify the target genes of NDRG2. We analyzed the Q-omics dataset and found that the EHF and NDRG2 gene expression was positively correlated in TNBC patients. In particular, the Pearson’s correlation coefficient (PCC) value of TNBC (0.4057) was higher than that of non-TNBC (0.3064) (Figure 1A,B). Consistent with the Q-omics dataset, there was a positive correlation between the endogenous expression of NDRG2 and EHF in the TNBC cell lines (Figure 1C). The PCC value was higher in the TNBC cell lines (0.57) than in the non-TNBC cell lines (0.2) (Figure 1D). The analysis of the TCGA dataset also showed that the expression of NDRG2 and EHF in breast cancer patients was positively correlated in the basal subtype (Appendix A). The PCC value was higher in the basal and triple negative (TN) subtypes (0.5) than in the basal subtype (0.48) (Appendix A). In contrast, the PCC values were relatively low in the luminal A (0.33), luminal B (0.21), and HER2 (0.06) subtypes (Appendix A). These data suggest that the NDRG2 and EHF expression levels are positively correlated in TNBC subtypes. In addition, the heatmap expression data from breast cancer patient biopsies showed that most of the NDRG2 and EHF expression levels were positively correlated (Figure 1E). On this basis, we investigated the expression levels of NDRG2 and EHF in TNBC cell lines, including MDA-MB-231, BT-549, Hs578T, and MDA-MB-453. These four TNBC cell lines each showed a positive correlation. The MDA-MB-453 cells had a high expression of both NDRG2 and EHF genes, whereas other cell lines have little expression of either gene (Figure 1F–H). To confirm the correlation between NDRG2 and EHF in the TNBC cell line, we used a previously established MDA-MB-231-NDRG2 cell line [24]. As expected, the EHF mRNA and protein levels were significantly increased in MDA-MB-231-NDRG2 cells, but not in wild-type and mock cells (Figure 1I–K). Consistent with these results, the stable overexpression of NDRG2 in 4T1 mouse mammary tumor cells resulted in an increased expression of EHF compared to mock cells (Appendix A). Collectively, these results suggest that, in TNBC cells, the EHF expression is strongly correlated with NDRG2 expression, which suppresses breast cancer progression.

### 3.2. EHF Inhibited Cell Growth and Promoted Paclitaxel-Induced Apoptosis

The correlation between EHF and NDRG2 led us to hypothesize that EHF may function as a tumor suppressor in TNBC cells. Recently, Sakamoto et al. demonstrated the tumor suppressor role of EHF in MDA-MB-231 cells [26]. Accordingly, we further investigated the functional role of EHF in breast cancer progression. As shown in Appendix A, the transient overexpression of EHF in MDA-MB-231 and MDA-MB-453 cells significantly inhibited the proliferation of both cell lines. Consistent with these data, the MDA-MB-231-EHF stable cells showed a reduced proliferation rate compared to mock cells (Figure 2A,B). These data suggest that EHF inhibits the growth of TNBC cells. We next evaluated the effect of EHF on apoptosis using paclitaxel, which is a commonly used chemotherapeutic drug for breast cancer. Paclitaxel treatment significantly increased the total apoptotic cells in MDA-MB-231-EHF cells compared to MDA-MB-231-mock cells at 24 and 48 h after treatment (Figure 2C,D). Consistent with these results, the transient overexpression of EHF also slightly increased the total apoptotic cells after paclitaxel treatment in MDA-MB-231 and MDA-MB-453 cells (Appendix A). These results suggest that EHF sensitizes MDA-MB-231 and MDA-MB-453 cancer cells to chemotherapeutic drugs. Collectively, these results demonstrated the tumor suppressive role of EHF by inhibiting growth and enhancing drug sensitivity in TNBC cells.

### 3.3. Induction of Senescence and Inhibition of Cell Migration by EHF Expression

Based on the significant growth retardation in MDA-MB-231-EHF cells, we hypothesized that EHF might be associated with senescence. Senescent cells are known to have a high activity of senescence-associated β-galactosidase (SA-β-Gal), and SA-β-Gal staining showed that MDA-MB-231-EHF cells had a high activity of SA-β-Gal compared to that in mock cells (Figure 3A). The number of SA-β-Gal positive cells was significantly higher in MDA-MB-231-EHF cells than in mock cells (Figure 3B). In addition, the expression of p21^WAF1/CIP1^ was increased in MDA-MB-231-EHF cells (Figure 3C). Next, we evaluated the effect of EHF on the migratory ability of TNBC cells. To rule out the effect of cell proliferation on the wound healing assay, the cells were treated with mitomycin C. The growth rate of MDA-MB-231-mock and MDA-MB-231-EHF cells was identically attenuated by mitomycin C treatment (Appendix A). The wound healing assay showed that the transient overexpression of EHF significantly inhibited the migration of MDA-MB-231 and MDA-MB-453 cells (Appendix A). Consistent with this finding, the Transwell assay with mitomycin C treatment showed that the stable overexpression of EHF significantly inhibited the migration and invasion ability of MDA-MB-231 cells (Figure 3D,E).

### 3.4. EHF Inhibited EMT in TNBC Cells

As the migratory ability is related to EMT [27], we investigated whether the EHF expression can affect EMT. A previous study showed that the EHF promotes the expression of E-cadherin by directly binding to the promoter region [28]. To investigate the effect of EHF on EMT-related genes, we analyzed the mRNA expression of EHF, E-cadherin, Snail, and Slug using the SEEK platform (Figure 4A). There was a strong positive correlation between EHF and E-cadherin, whereas Snail and Slug showed a weak inverse correlation with EHF. Consistent with this information, the Q-omics dataset analysis of the TNBC cell line showed a significant positive correlation between the EHF and E-cadherin mRNA expression levels and a negative correlation with Slug (Figure 4B). In addition, morphological changes were observed in stable MDA-MB-231-EHF cells (Figure 4C). Stable MDA-MB-231-EHF cells had increased cell-to-cell contact compared to that with mock cells. The mesenchymal spindle-shaped morphology of wild-type cells was changed to an elongated and polygonal epithelial cell morphology upon stable EHF overexpression, reinforcing that EHF could induce mesenchymal–epithelial transition. Next, we evaluated the protein expression of the EMT markers. Indeed, E-cadherin expression was increased in the stable EHF-overexpressing cells, and Snail expression was downregulated compared to that in mock cells (Figure 4D). Consistent with this finding, the transient overexpression of EHF reduced the Slug expression, and promoted E-cadherin expression in MDA-MB-231 cells (Figure 4E). In MDA-MB-453 cells, the transient overexpression of EHF reduced the expression of Snail and N-cadherin (Figure 4F). Furthermore, the knockdown of EHF using specific siRNA significantly increased the mRNA expression of N-cadherin and Slug in MDA-MB-231-EHF cells (Figure 4G). Taken together, these data suggest that EHF exerts antitumor effects by suppressing the EMT, migration, and invasion of TNBC cells.

### 3.5. EHF Overexpression Inhibited the Phosphorylation of STAT3

STAT3 is considered a target for anticancer therapy since it is associated with tumor initiation and progression [29]. A previous study showed that EHF suppresses STAT3 activation in prostate cancer cells [18]. Accordingly, we investigated whether the EHF expression is correlated with the suppression of the STAT3 signaling pathway in breast cancer cells. As expected, EHF overexpression suppressed STAT3 phosphorylation in MDA-MB-231 and MDA-MB-453 cells (Figure 5A,B). To confirm whether EHF suppresses STAT3 signaling, we used two specific siRNAs against EHF in MDA-MB-231-NDRG2 cells, which have a high expression of EHF. Although the expression level of phospho-STAT3 was low in MDA-MB-231-NDRG2 cells, the knockdown of EHF rescued STAT3 phosphorylation in a time-dependent manner (Figure 5C). Furthermore, we found that STAT3 signaling was suppressed in MDA-MB-231-EHF stable cells compared to mock cells (Figure 5D). These data suggest that EHF suppresses STAT3 activation in TNBC cells.

### 3.6. EHF Exhibited Antitumor Effects in Mouse Mammary Tumor Cells

To further evaluate the role of EHF in tumor progression in vivo, we established a stable 4T1-EHF cell line (Figure 6A,B). Consistent with previous data, the overexpression of EHF significantly attenuated the proliferation rate of 4T1 cells (Figure 6C). 4T1-EHF cells had high SA-β-Gal activity compared to that in mock cells (Figure 6D). In addition, the p21^WAF1/CIP1^ expression was also upregulated in EHF-overexpressing 4T1 cells (Figure 6E). These data suggest that EHF also inhibits proliferation and induces the cellular senescence of mouse mammary tumor cells. Next, we examined the effect of EHF on migration ability. To rule out the effect of cell growth, cells were treated with mitomycin C. The proliferation of mock and EHF cells was similarly attenuated by treatment with mitomycin C (Appendix A). Compared to mock cells, the 4T1-EHF cells exhibited reduced migration and invasion ability in the wound healing assay and Transwell assay (Figure 6F–H). In addition, the doxorubicin treatment resulted in a significantly higher apoptotic rate in 4T1-EHF cells than in 4T1-mock cells (Figure 6I,J).

### 3.7. EHF Suppressed Breast Cancer Progression

We next investigated whether the EHF overexpression could inhibit tumor progression in vivo. Balb/c mice were injected subcutaneously with 4T1-mock or 4T1-EHF cells (Figure 7A). The tumor volume resulting from the subcutaneous injection of 4T1-mock or 4T1-EHF cells was significantly lower in 4T1-EHF tumors (Figure 7B). Interestingly, two of the 4T1-EHF tumor-bearing mice showed complete tumor regression within 21 days (Appendix A). The tumor weight was also significantly lower in 4T1-EHF tumors (Figure 7C). Consistent with the MDA-MB-231 data, STAT3 signaling was suppressed, and its Slug expression was also downregulated in tumor cells derived from 4T1-EHF tumor tissue (Figure 7D). Next, we investigated whether the EHF overexpression in 4T1 cells affected splenomegaly induced by tumor cells. The spleen length and weight in tumor-bearing mice were increased in 4T1-mock, compared to PBS control mice. However, 4T1-EHF tumor-bearing mice exhibited a reduced spleen weight (Figure 7E,F). Next, to examine the effect of EHF on the migration ability in vivo, the number of pulmonary nodules formed by the subcutaneous injection of tumor cells was evaluated. The EHF overexpression suppressed the migration of tumor cells, so the number of pulmonary nodules was significantly reduced compared to that in mock tumors (Appendix A). To confirm the effect of EHF on migration ability, we intravenously injected 4T1-mock or 4T1-EHF cells into Balb/c mice (Figure 7G). At 2 weeks post-injection, we counted the number of pulmonary nodules resulting from the migration of intravenously injected 4T1 cells. The number of nodules was significantly reduced in 4T1-EHF tumor-bearing mice compared to 4T1-mock tumor-bearing mice (Figure 7H). These data suggest that EHF is a tumor suppressor gene that suppresses the proliferation, EMT, and migration ability of tumor cells in vivo, in agreement with previous data. To further investigate whether EHF induces senescence at the tumor site, tumor tissue sections were stained for SA-β-Gal activity. Although there was no SA-β-Gal activity in the mock tumor tissue, a portion of the EHF tumor tissue showed SA-β-Gal activity (Figure 7I). Taken together, the results show that EHF suppressed tumor progression by inducing cellular senescence in vivo.

### 3.8. Breast Cancer Patients with High EHF Expression Showed an Increased Survival Rate

To evaluate the effect of EHF on the prognosis of breast cancer patients, we analyzed the survival rate of TNBC patients according to EHF expression using Kaplan–Meier plotter. TNBC patients with a high EHF expression had a longer relapse-free survival rate and overall survival rate than those of patients with a low EHF expression (Figure 8A,B). These data suggest that a low EHF expression is associated with a poor prognosis of TNBC. In addition, the high expression of EHF in breast cancer patients receiving chemotherapy showed prolonged relapse-free survival (Figure 8C). The disease-free interval was significantly prolonged in patients with a high EHF expression (Figure 8D). The pathological TNM staging system refers to the size of the tumor (T), the extent of spread to the lymph nodes (N), and the presence of metastasis (M). We also found that the high pathological T and N stages were associated with a low EHF expression in breast cancer patients when their EHF expression was divided by the median value (Figure 8E). The difference in the pathological T and N stages between patients with high and low expressions of EHF was more obvious when they were divided by tertile value, highlighting the clinical importance of EHF in breast cancer patients (Figure 8F). These data revealed that EHF is a potential prognostic marker in breast cancer patients.

## 4. Discussion

In the current study, we investigated the tumor suppressive role of EHF, which is thought to inhibit migration and invasion through the induction of senescence and the inhibition of the STAT3 signaling pathway in breast cancer cells. EHF is a context-dependent transcription factor, as it has tumor-promoting role in gastric and thyroid cancer [16,30]. In contrast, several studies have shown that EHF functions as a tumor suppressor gene in pancreatic and prostate cancer [28,31]. However, what determines the function of EHF in various cancers remains unclear. Intriguingly, a previous study investigated the role of EHF in esophageal squamous cell carcinoma (ESCC) and found that the altered subcellular location of EHF resulted in ESCC [32]. Whereas EHF in normal esophageal epithelium is expressed in the nucleus, EHF in ESCC is expressed in the cytoplasm. This suggests that the subcellular location of EHF may determine the role of EHF in cancer cells. However, the functional role of EHF in breast cancer remains unclear. We found that the EHF expression was induced by the transient overexpression of NDRG2 in MDA-MB-231 and MDA-MB-453 cells (Appendix A). In addition, the high expression of EHF was observed in MDA-MB-231-NDRG2 cells, which was previously established in our recent study [12]. These data suggest that NDRG2 may be involved in the induction of EHF expression in TNBC cells. However, how NDRG2 regulates EHF expression needs to be further investigated as NDRG2 is not a transcription factor. Presumably, a transcription factor that is regulated by NDRG2 may induce the EHF expression. Interestingly, our recent study showed that NDRG2 overexpression increased the phosphorylation of p38, which consequently phosphorylates the substantial portions of the ETS transcription factor family members, altering their transcriptional functions and regulating their activity [33,34]. Specifically, the phosphorylation of Elk-1 by MAPK leads to interaction with p300, yielding Elk-1-p300 complexes, which play a crucial role in gene activation and chromatin remodeling [35]. Therefore, p38 MAPK might be involved in the induction of EHF expression by NDRG2. However, since stable MDA-MB-231-EHF cells also had a high expression of NDRG2 (Appendix A), there might be a positive feedback loop between EHF and NDRG2. Therefore, further studies are required to elucidate the relationship between EHF and NDRG2.

Recently, Albino et al. demonstrated that EHF represses the STAT3 signaling pathway by directly binding to the promoter region of IL-6 and downregulating its expression in prostate cancer cells [31]. Similarly, we observed that EHF inhibits the STAT3 signaling pathway, and the knockdown of EHF by siRNA rescues STAT3 activation in TNBC cells. However, the molecular mechanisms responsible for the EHF-mediated inhibition of the STAT3 pathway in TNBC cells might be different from those in prostate cancer cells because the mRNA level of IL-6 was significantly increased by EHF overexpression in MDA-MB-453 cells (Appendix A). This result suggests that EHF suppresses STAT3 activation through an alternative pathway, but not the IL-6/STAT3 axis.

Senescence can be induced by various tumor suppressor genes. Among them, the p53 and pRB proteins are crucial proteins that induce cellular senescence. They regulate the expression of genes that are related to cell cycle arrest and maintain growth arrest of senescent cells in response to various stimuli [36]. Similarly, EHF might be associated with the expression of several genes that regulate senescence, including p21^WAF1/CIP1^. p21^WAF1/CIP1^ is a key effector downstream of p53, but p21^WAF1/CIP1^ expression could be increased independently of p53. TNBC cell lines such as MDA-MB-231 and MDA-MB-453 are known to have mutations in p53. Therefore, there might be a mechanism by which EHF regulates the p21^WAF1/CIP1^ expression independently of p53. The precise mechanism by which EHF regulates cellular senescence and p21^WAF1/CIP1^ in breast cancer needs to be investigated.

We have also shown that EHF inhibits EMT, which is involved in the morphological changes of MDA-MB-231-EHF cells. A recent study showed that EHF upregulates the E-cadherin expression by directly binding to its promoter region in pancreatic cancer cells [28]. Similarly, in our current study, we demonstrated a simultaneous increase in E-cadherin and EHF in TNBC cells. Thus, the EHF-mediated increase in E-cadherin might be the result of transcriptional regulation in the promoter region of E-cadherin by EHF. More detailed studies are warranted to investigate the precise mechanism of EMT inhibition by the EHF gene.

Kaplan–Meier plotter analysis showed that TNBC patients with a high expression of EHF had a longer relapse-free survival rate. Our study highlights the importance of EHF as a prognostic marker of TNBC. Therefore, monitoring the expression of EHF in TNBC patients might be useful for identifying patients with a high risk of relapse and low overall survival. A previous study showed that a low expression of EHF correlates with the promoter methylation in prostate and pancreatic cancer [28,32]. In these cases, demethylating agents led to the re-expression of EHF and suppressed cancer progression. However, in TNBC patients with low EHF expression, the re-induction or modulation of EHF expression by demethylating agents needs to be considered with caution because demethylating agents have broad transcriptional effects and EHF has a tumor promoting role in gastric and thyroid cancers. Therefore, the effect of EHF overexpression in vivo needs to be investigated in detail.

In conclusion, as summarized in Figure 8G, this study demonstrates the anticancer effect of EHF via its ability to suppress STAT3 signaling and induce senescence, suggesting that EHF is a potent tumor suppressor gene and a prognostic marker in TNBC cells. Further investigation into the mechanism of EHF action and its correlation with NDRG2 expression may provide insights into the development of specific therapies for TNBC patients.

## 5. Conclusions

In conclusion, this study suggested that EHF could be a novel prognostic marker in TNBC and demonstrated that EHF overexpression not only promotes tumor cell apoptosis and cellular senescence but also suppresses tumor cell migration, invasion, and proliferation, which might be mediated by the STAT3 signaling pathway in TNBC cells. Therefore, EHF may be a novel tumor suppressor gene that acts by inducing senescence in breast cancer (Figure 8G).

## Figures and Tables

**Figure 1 cancers-15-05270-f001:**
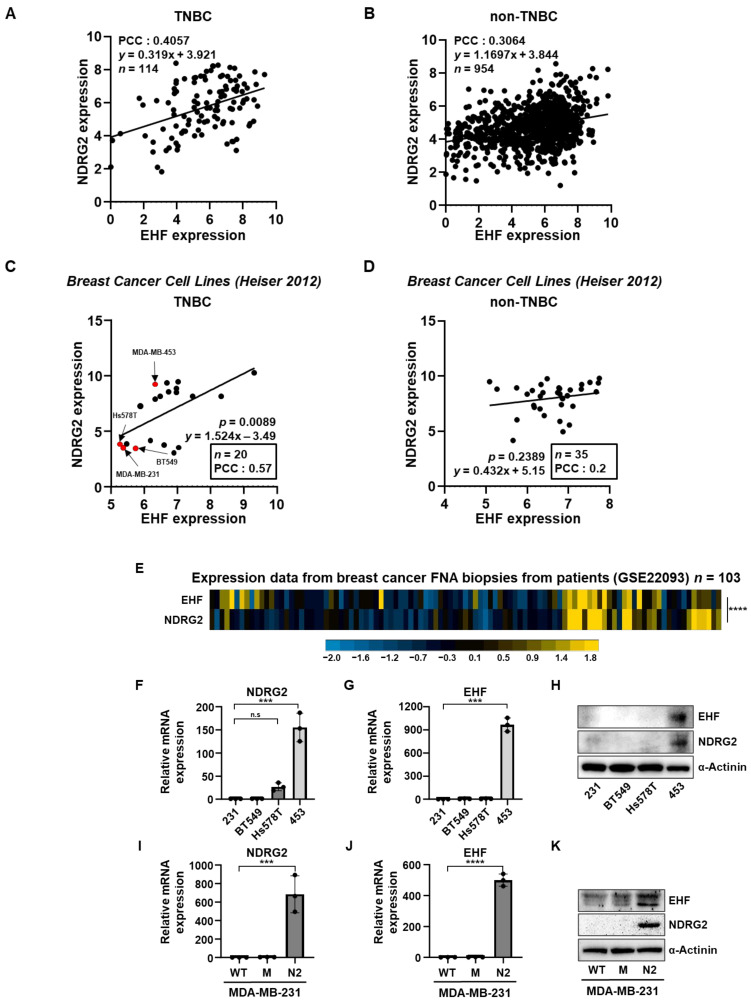
The positive correlation between the NDRG2 and EHF expression levels. (**A**,**B**) Correlation analysis of the NDRG2 and EHF expression of TNBC and non-TNBC patients. (**C**,**D**) Correlation analysis of NDRG2 and EHF expression of TNBC and non-TNBC breast cancer cell lines. Red dots indicate the MDA-MB-453, MDA-MB-231, BT549 and Hs578T respectively. (**E**) Heatmap showing the correlation of NDRG2 and EHF expression from the FNA biopsies of breast cancer patients. (**F**,**G**) RT-qPCR data showing the relative expression of NDRG2 and EHF in MDA-MB-231, BT-549, Hs578T, and MDA-MB-453 cells. (**H**) Western blot assay showing EHF and NDRG2 protein levels in MDA-MB-231, BT-549, Hs578T, and MDA-MB-453 cells. (**I**,**J**) RT-qPCR data showing the relative expression of NDRG2 and EHF in MDA-MB-231-wild type, -mock, and -NDRG2 cells. (**K**) Western blot assay showing EHF and NDRG2 protein levels of MDA-MB-231 wild type, -mock, -NDRG2 cells. *** *p* < 0.001, **** *p* < 0.0001.

**Figure 2 cancers-15-05270-f002:**
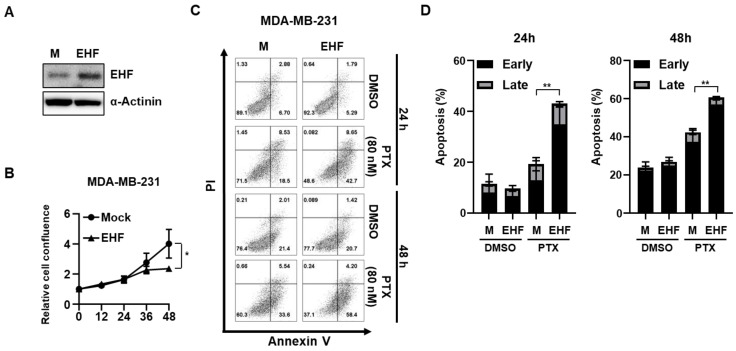
EHF inhibited cell growth and promoted paclitaxel-induced apoptosis. (**A**) Western blot assay showing EHF protein levels in MDA-MB-231-mock and -EHF cells. (**B**) Relative cell confluence of MDA-MB-231-mock and -EHF cells was calculated using IncuCyte ZOOM^®^. Circle symbol represents MDA-MB-231-mock and triangle symbol represents MDA-MB-231-EHF. (**C**) MDA-MB-231-mock and -EHF cells were treated with DMSO or 80 nM of paclitaxel for 24 or 48 h. Cells were stained for Annexin V/PI analysis. (**D**) Early and late apoptosis rate of MDA-MB-231-mock and -EHF cells. * *p* < 0.05, ** *p* < 0.01.

**Figure 3 cancers-15-05270-f003:**
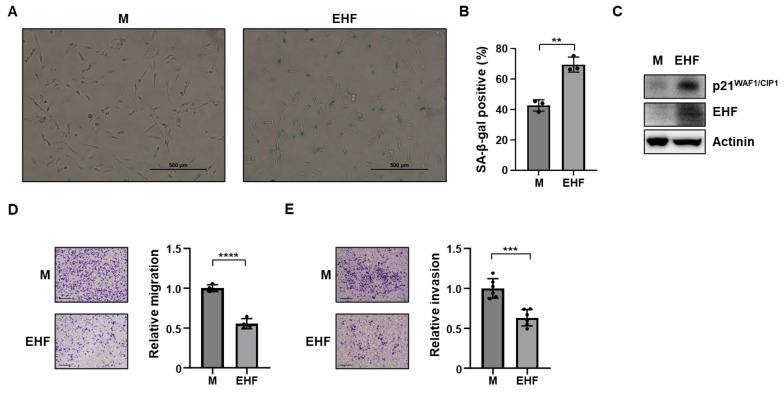
EHF overexpression induced tumor cell senescence, and inhibited migration. (**A**) MDA-MB-231-mock and -EHF cells were subjected to SA-β-Gal assay. SA-β-Gal staining (10× magnification) is shown in brightfield images. Scale bar indicates 500 μm. (**B**) Graph showing the percentage of SA-β-Gal positive cells. (**C**) Western blot assay showing p21^WAF1/CIP1^ and EHF protein levels of MDA-MB-231-mock and -EHF cells. (**D**) Images of migrated MDA-MB-231-mock and -EHF cells (left). A graph showing the relative migration of MDA-MB-231-mock and -EHF cells (right). (**E**) Images of invaded MDA-MB-231-mock and -EHF cells (left). A graph showing the relative invasion of MDA-MB-231-mock and -EHF cells (right). Scale bar indicates 300 μm. ** *p* < 0.01, *** *p* < 0.001, **** *p* < 0.0001.

**Figure 4 cancers-15-05270-f004:**
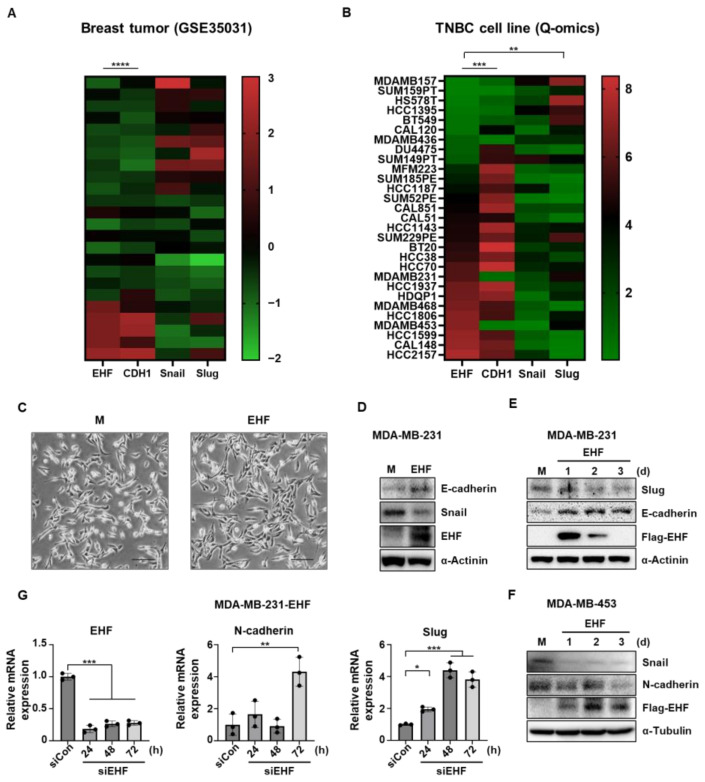
Negative correlation of EHF expression with the expression of EMT marker genes. (**A**) Heatmap showing the relative expression of EHF, CDH1, SNAI1, and SNAI2 in breast tumor. (**B**) Heatmap showing the relative expression of EHF, CDH1, SNAI1, and SNAI2 in TNBC cell lines. (**C**) Brightfield images of MDA-MB-231-mock and -EHF cells. Scale bar indicates 500 μm. (**D**) Western blot assay showing E-cadherin, Snail, and the EHF protein levels of MDA-MB-231-mock and -EHF cells. (**E**) Western blot assay showing Slug, E-cadherin, and Flag-EHF protein levels in MDA-MB-231 cells after transfection with pCMV-Taq2B (mock) and pCMV6-Entry-EHF (EHF). (**F**) Western blot assay showing Snail, N-cadherin, and Flag-EHF protein levels in MDA-MB-453 cells after transfection with pCMV-Taq2B (mock) and pCMV6-Entry-EHF (EHF). (**G**) RT-qPCR data showing the relative expression of EHF, N-cadherin, and Slug in MDA-MB-231 cells after transfection of control siRNA and siRNA against EHF. * *p* < 0.05, ** *p* < 0.01, *** *p* < 0.001, **** *p* < 0.0001.

**Figure 5 cancers-15-05270-f005:**
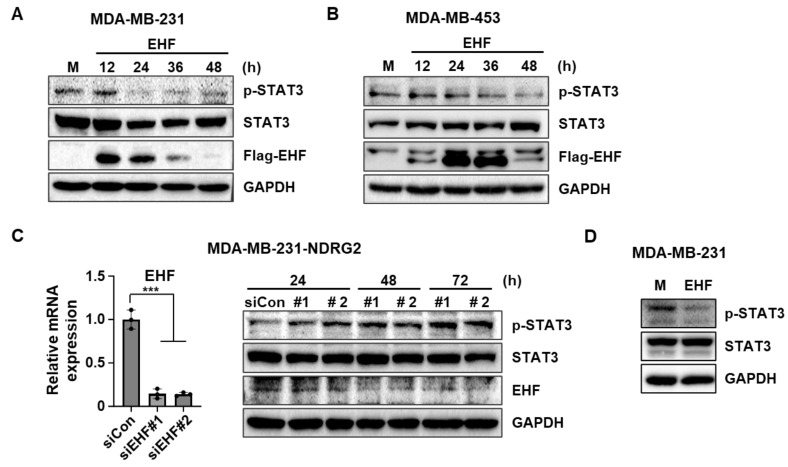
EHF suppressed EMT through the STAT3 signaling pathway. (**A**) Western blot assay showing p-STAT3, STAT3, and Flag-EHF protein levels of MDA-MB-231 cells after transfection with pCMV-Taq2B (mock) and pCMV6-Entry-EHF (EHF). (**B**) Western blot assay showing p-STAT3, STAT3, and Flag-EHF protein levels in MDA-MB-453 cells after transfection with pCMV-Taq2B (mock) and pCMV6-Entry-EHF (EHF). (**C**) RT-qPCR data showing a relative EHF expression in MDA-MB-231-NDRG2 cells after transfection with control siRNA and two siRNAs (number 1 (#1) and number 2 (#2)) against EHF (left). Western blot assay showing p-STAT3, STAT3, and EHF protein levels of MDA-MB-231-NDRG2 cells after the transfection of control siRNA and two siRNAs against EHF (right). (**D**) Western blot assay showing p-STAT3 and STAT3 protein levels in MDA-MB-231-mock and -EHF cells. *** *p* < 0.001.

**Figure 6 cancers-15-05270-f006:**
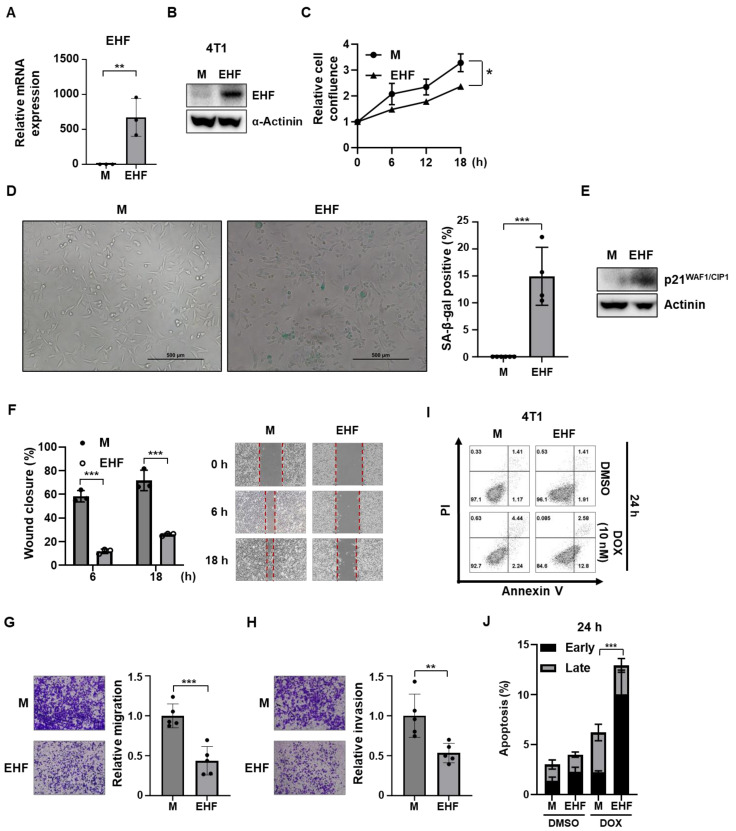
Overexpression of EHF induced senescence, apoptosis, and suppressed migration of 4T1 cells. (**A**) RT-qPCR data showing the relative EHF expression of 4T1-mock and -EHF cells. (**B**) Western blot assay showing EHF protein levels of 4T1-mock and -EHF cells. (**C**) Relative cell confluence of 4T1-mock and -EHF cells was calculated using IncuCyte ZOOM^®^. Circle symbol represents 4T1-mock and triangle symbol represents 4T1-EHF. (**D**) 4T1-mock and -EHF cells were subjected to SA-β-Gal assay. SA-β-Gal staining (10X magnification) is shown in brightfield images (left). Graph showing percentage of SA-β-Gal-positive cells (right). Scale bar indicates 500 μm. (**E**) Western blot assay showing p21^WAF1/CIP1^ protein levels in 4T1-mock and -EHF cells. (**F**) Wound closure percentage was calculated as the ratio of the remaining gap at the time point to the original scratch gap at 0 h (left). Representative images of wound healing analysis of 4T1-mock and -EHF cells (right). Red dotted lines indicate the borders of the wound at 0, 6, 18 h post scratch. (**G**) Representative images of migrated 4T1-mock and -EHF cells (left). A graph showing the relative migration of 4T1-mock and -EHF cells (right). (**H**) Representative images of invaded 4T1-mock and -EHF cells (left). A graph showing the relative invasion of 4T1-mock and -EHF cells (right). Scale bar indicates 300 μm. (**I**) 4T1-mock and -EHF cells were treated with DMSO or 10 nM of doxorubicin for 24 h. Cells were stained for Annexin V/PI analysis. (**J**) Early and late apoptosis rate of 4T1-mock and -EHF cells. * *p* < 0.05, ** *p* < 0.01 *** *p* < 0.001.

**Figure 7 cancers-15-05270-f007:**
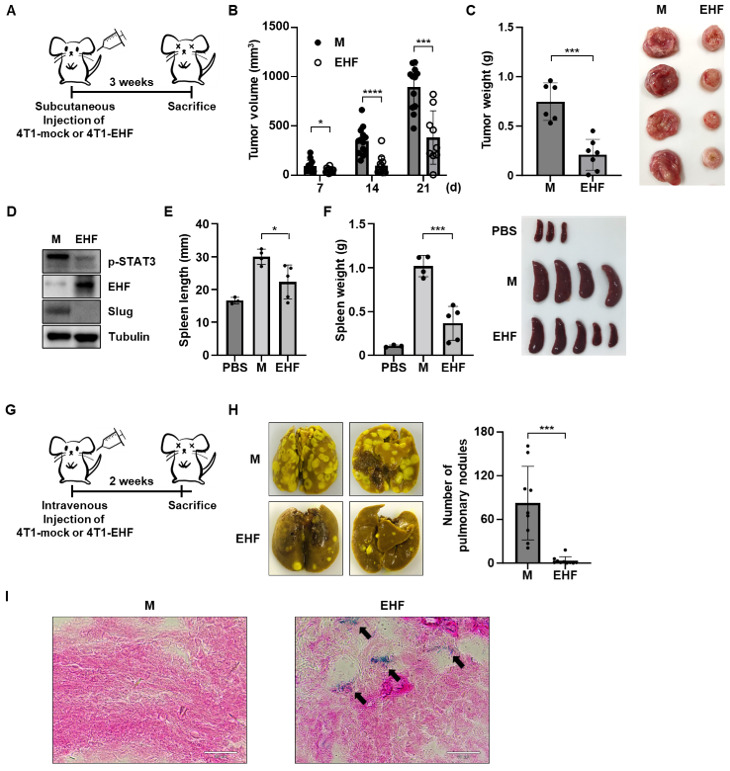
Overexpression of EHF inhibited tumor progression in vivo. (**A**) Schematic representation of the experimental protocol. (**B**) Tumor volume was measured in Balb/c mice that received the subcutaneous injection of 4T1-mock and -EHF cells. (**C**) Tumor weight was measured in Balb/c mice that are injected subcutaneously with 4T1-mock and -EHF cells (left). Representative images of tumors (right). (**D**) Western blot assay showing p-STAT3, EHF, and Slug protein levels in tumor cells derived from 4T1-mock and 4T1-EHF tumors. (**E**) Spleen length in Balb/c mice injected subcutaneously with 4T1-mock and -EHF cells. (**F**) Spleen weight in Balb/c mice injected subcutaneously with 4T1-mock and -EHF cells (left). Representative images of spleens (right). (**G**) Schematic diagram of the experimental protocol. (**H**) Representative images of the lungs in the Balb/c mouse that received intravenous injection of 4T1-mock and -EHF cells (left). A graph showing the number of pulmonary nodules (right). (**I**) Representative images showing the SA-β-Gal staining of tumors from Balb/c mice injected subcutaneously with 4T1-mock and -EHF cells. Arrow indicates SA-β-Gal positive tumor sections. Scale bar indicates 30 μm. * *p* < 0.05, *** *p* < 0.001, **** *p* < 0.0001.

**Figure 8 cancers-15-05270-f008:**
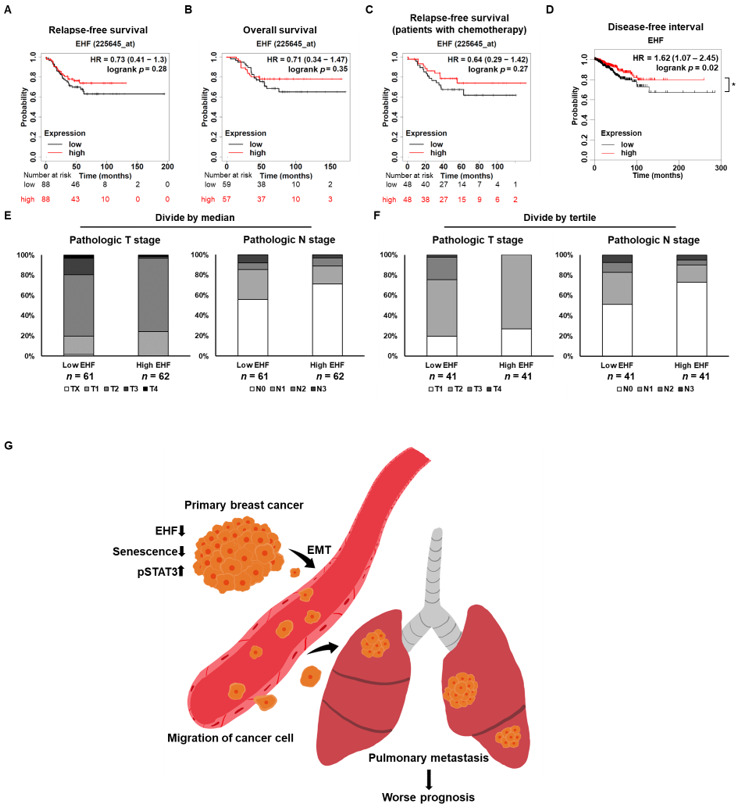
EHF as a prognostic marker of TNBC patients. (**A**,**B**) The relapse-free survival and overall survival plot according to the expression level of EHF in TNBC patients using the Kaplan–Meier plotter. Patients were divided by the median value of EHF and the red line indicates a high EHF expression. (**C**) Relapse-free survival of TNBC patients treated with chemotherapy using the Kaplan–Meier plotter. (**D**) Disease-free interval of TCGA breast cancer patients. (**E**) A graph showing the proportion of pathological T stage (left) and N stage (right) according to the EHF expression level that is divided by the median value. (**F**) A graph showing the proportion of pathological T stage (left) and N stage (right) according to the EHF expression level that is divided by the tertile value. (**G**) Schematic diagram of how downregulation of EHF contributes to STAT3 signaling activation and EMT, whereas it inhibits senescence, leading to migration of tumor cells, and resulting in worse prognosis. * *p* < 0.05.

**Table 1 cancers-15-05270-t001:** Quantitative PCR primer sequences.

Gene	Species	Sequence
GAPDH	Human	FW—CTGGGCTACACTGAGCACCA RV—CCAGCGTCAAAGGTGGAG
EHF	Human	FW—TGCAGCATCTGAAGTGGAAC RV—AGGAAGGTGACTGGTGGTTG
NDRG2	Human	FW—TCTGTCACTTTCACTGTCTAR RV—CCAGAGATGGGTACTGATAT
N-CADHERIN	Human	FW—CCTCCAGAGTTTACTGCCATGAC RV—GTAGGATCTCCGCCACTGATTC
SLUG	Human	FW—ATCTGCGGCAAGGCGTTTTCCA RV—GAGCCCTCAGATTTGACCTGTC
CYCLOPHILIN	Mouse	FW—CATACAGGTCCTGGCATCTTGTC RV—AGACCACATGCTTGCCATCCAG
EHF	Mouse	FW—ATGCAATGTTTCCAGCGGTT RV—GTCGAACTCCTGGAAAGGGA
NDRG2	Mouse	FW—CAGCATCACAGGGCACTTGA RV—CGCCGAGACCTGAACTTTG

## Data Availability

The data presented in this study are available on request from the corresponding author.

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
