# Peer review of "Anticancer Effect of E26 Transformation-Specific Homologous Factor through the Induction of Senescence and the Inhibition of Epithelial–Mesenchymal Transition in Triple-Negative Breast Cancer Cells"

_cancers, 2023, doi:10.3390/cancers15215270_

Round 1
Reviewer 1 Report
Comments and Suggestions for Authors
Lim et al evaluated the effect of ETS homologous factor (EHF) in breast cancer. Authors identified the correlation between NDRG2 and EHF expression in TNBC patients and cell lines. Using knockdown and overexpression approaches, authors demonstrated that EHF expression is associated with growth retardation, apoptosis and senescence as well as inhibition of EMT and migration. EHF inhibited STAT3 phosphorylation. In addition, in vivo studies using syngeneic mouse breast cancer cells, EHF was associated with reduced tumor growth and metastasis. In patients, it was associated with a longer survival. The manuscript was well written. methods and results were clearly described. Conclusions are supported by results. there are multiple layers of evidence.
Author Response
Reviewer #1
Lim et al evaluated the effect of ETS homologous factor (EHF) in breast cancer. Authors identified the correlation between NDRG2 and EHF expression in TNBC patients and cell lines. Using knockdown and overexpression approaches, authors demonstrated that EHF expression is associated with growth retardation, apoptosis and senescence as well as inhibition of EMT and migration. EHF inhibited STAT3 phosphorylation. In addition, in vivo studies using syngeneic mouse breast cancer cells, EHF was associated with reduced tumor growth and metastasis. In patients, it was associated with a longer survival. The manuscript was well written. Methods and results were clearly described. Conclusions are supported by results. There are multiple layers of evidence.
- Thank you so much for the reviewer’s comment.
Reviewer 2 Report
Comments and Suggestions for Authors
Here, the authors present a study on the anticancer effect of EHF in triple-negative breast cancer (TNBC) cells. The authors found that EHF, a member of the ETS transcription factor family, is positively correlated with NDRG2, a tumor suppressor gene, in TNBC cells and patients. The study showed that EHF overexpression inhibited cell growth, induced senescence, and suppressed epithelial-mesenchymal transition (EMT) and STAT3 signaling in TNBC cells. The study also demonstrated that EHF overexpression reduced tumor size and lung metastasis in a mouse model of TNBC. Finally, the study suggested that EHF might be a tumor suppressor gene and a prognostic marker for TNBC.
Main points:
The authors provide evidence for the role of EHF in TNBC using various methods, such as gene expression analysis, cell culture experiments, and animal models.
The study reveals a novel mechanism of EHF-mediated senescence and EMT regulation through STAT3 inhibition in TNBC cells.
This work has potential implications for the development of new therapeutic strategies for TNBC based on EHF expression or activity.
Minor points:
The study does not explain how EHF expression is regulated by NDRG2 or other factors in TNBC cells. This should at least be addressed in the discussion.
The study does not compare the effect of EHF on other breast cancer subtypes or other types of cancer. This should be discussed as several works investigated the role of EHF in gastric cancer, or various carcinomas.
The authors do not address the possible side effects or limitations of EHF overexpression or modulation in vivo.
Comments on the Quality of English Language
Fine. Minor mistakes to be corrected
Author Response
Reviewer #2
Here, the authors present a study on the anticancer effect of EHF in triple-negative breast cancer (TNBC) cells. The authors found that EHF, a member of the ETS transcription factor family, is positively correlated with NDRG2, a tumor suppressor gene, in TNBC cells and patients. The study showed that EHF overexpression inhibited cell growth, induced senescence, and suppressed epithelial-mesenchymal transition (EMT) and STAT3 signaling in TNBC cells. The study also demonstrated that EHF overexpression reduced tumor size and lung metastasis in a mouse model of TNBC. Finally, the study suggested that EHF might be a tumor suppressor gene and a prognostic marker for TNBC.
Main points:
The authors provide evidence for the role of EHF in TNBC using various methods, such as gene expression analysis, cell culture experiments, and animal models.
The study reveals a novel mechanism of EHF-mediated senescence and EMT regulation through STAT3 inhibition in TNBC cells.
This work has potential implications for the development of new therapeutic strategies for TNBC based on EHF expression or activity.
- We thank you for the reviewer’s comment on our experiment.
Minor points:
The study does not explain how EHF expression is regulated by NDRG2 or other factors in TNBC cells. This should at least be addressed in the discussion.
- Thank you so much for the reviewer’s detailed comment. Whether EHF is regulated by NDRG2 in breast cancer cells is still unclear. However, we found that EHF expression is induced by transient transfection of NDRG2 in MDA-MB-231 and MDA-MB-453 cells (data not shown). We think how NDRG2 regulates EHF in breast cancer needs further investigation, because NDRG2 is not a transcription factor. Presumably, a transcription factor that is regulated by NDRG2 may be involved in the induction of EHF in breast cancer. Accordingly, we have added and revised the description in the discussion section.
The study does not compare the effect of EHF on other breast cancer subtypes or other types of cancer. This should be discussed as several works investigated the role of EHF in gastric cancer, or various carcinomas.
- We thank you for the reviewer’s comment on experimental design. As mentioned in the manuscript, while EHF functions as tumor suppressor in prostate and pancreatic cancer, it functions as oncogene in gastric and thyroid cancer. We think that the mechanism by which EHF functions in cancer cells needs to be further investigated. It is possible that the subcellular localization of EHF may determine its function, as EHF expression in the cytoplasm led to the esophageal squamous cell carcinoma (ESCC), whereas normal esophageal epithelium has EHF expression in the nucleus. In response to the reviewer’s comment, we have revised the discussion section.
The authors do not address the possible side effects or limitations of EHF overexpression or modulation in vivo.
- As EHF is a context-dependent transcription factor, overexpression of EHF in vivo may lead to tumor development, as observed in gastric and thyroid cancer. Therefore, modulation or upregulation of EHF expression in vivo should be considered with caution, and it requires further investigation. In response to the reviewer’s comment, the discussion section on this topic has been revised.
Reviewer 3 Report
Comments and Suggestions for Authors
Breast cancer is a common malignancy worldwide and is associated with significant morbidity and mortality. The ETS homologous factor (EHF), a transcription factor belonging to the ETS family, has been implicated in various biological processes, including cell proliferation, differentiation, and apoptosis. However, its role in breast cancer progression remains unclear. The authors found that the tumor suppressor NDRG2 was correlated with EHF gene expression in triple-negative breast cancer cells. Moreover, overexpression of EHF resulted in reduced cell proliferation and enhanced apoptosis upon treatment with chemotherapeutic agents. Additionally, overexpression of EHF led to increased senescence-associated β-galactosidase activity and p21WAF1/CIP1 expression, suggesting that EHF may induce cellular senescence. Furthermore, they found that overexpression of EHF reduced the migratory ability and inhibited epithelial-mesenchymal transition (EMT). Moreover, EHF inhibited the phosphorylation of STAT3, a signaling molecule implicated in tumor progression. In vivo experiments showed that EHF overexpression reduced tumor size and lung metastasis. At the tumor site, there was an increase in β-galactosidase activity due to EHF overexpression. Finally, Kaplan-Meier analysis demonstrated that triple-negative breast cancer patients with high expression of EHF had a longer relapse-free survival rate. Taken together, their findings suggest that EHF plays a role in inhibiting breast tumor progression in triple-negative breast cancer cells by inducing senescence and regulating EMT and these results provide new insights into the mechanisms underlying breast cancer progression and may have clinical implications for the development of new therapeutic strategies targeting EHF in patients with triple-negative breast cancer. But I have several following concerns:
1. Why in Figure 1H, the immunoblot of MDA-MB-231 cells showed almost no expression of EHF, while in Figure 2A, the mock group expressed quite a lot of EHF?
2. In line 334, is EHF inducing EMT or inhibiting EMT?
3. The genes and proteins appearing in 3.4 should be properly described.
4. In Figure 4E, what does the immunoblot in the last line represent?
5. In Figure 7I, it is not clear how much the scale bar represents the display.
6. Please add a scale bar to the light microscopy and fluorescence microscopy in Figure 3.
7. Abbreviations should be defined When thery appears in the first time.
8. Please unify the format of references in the article, including the author's name, the case of words in the title of the article, the writing of the name of the journal, and the page number.
Author Response
Reviewer #3
Breast cancer is a common malignancy worldwide and is associated with significant morbidity and mortality. The ETS homologous factor (EHF), a transcription factor belonging to the ETS family, has been implicated in various biological processes, including cell proliferation, differentiation, and apoptosis. However, its role in breast cancer progression remains unclear. The authors found that the tumor suppressor NDRG2 was correlated with EHF gene expression in triple-negative breast cancer cells. Moreover, overexpression of EHF resulted in reduced cell proliferation and enhanced apoptosis upon treatment with chemotherapeutic agents. Additionally, overexpression of EHF led to increased senescence-associated β-galactosidase activity and p21WAF1/CIP1 expression, suggesting that EHF may induce cellular senescence. Furthermore, they found that overexpression of EHF reduced the migratory ability and inhibited epithelial-mesenchymal transition (EMT). Moreover, EHF inhibited the phosphorylation of STAT3, a signaling molecule implicated in tumor progression. In vivo experiments showed that EHF overexpression reduced tumor size and lung metastasis. At the tumor site, there was an increase in β-galactosidase activity due to EHF overexpression. Finally, Kaplan-Meier analysis demonstrated that triple-negative breast cancer patients with high expression of EHF had a longer relapse-free survival rate. Taken together, their findings suggest that EHF plays a role in inhibiting breast tumor progression in triple-negative breast cancer cells by inducing senescence and regulating EMT and these results provide new insights into the mechanisms underlying breast cancer progression and may have clinical implications for the development of new therapeutic strategies targeting EHF in patients with triple-negative breast cancer. But I have several following concerns:
- Why in Figure 1H, the immunoblot of MDA-MB-231 cells showed almost no expression of EHF, while in Figure 2A, the mock group expressed quite a lot of EHF?
- Thank you for the reviewer’s comment. In Figure 1H, it looks like that MDA-MB-231 cells have almost no expression of EHF because MDA-MB-453 cells have a very high expression of EHF. In fact, as shown in Figure 2A and 4B, MDA-MB-231-wt, and -mock cells show some EHF expression. However, compared to MDA-MB-453 cells, MDA-MB-231 cells appear to have relatively low expression of EHF.
- In line 334, is EHF inducing EMT or inhibiting EMT?
- We consistently showed in this manuscript that EHF inhibits EMT. Specifically, in breast cancer cells, we also observed that EHF overexpression caused the cells to switch from a mesenchymal state to an epithelial state (MET).
- The genes and proteins appearing in 3.4 should be properly described.
- We apologize for the confusion between gene and proteins in section 3.4. According to the reviewer’s comment, we have revised some sentences in section 3.4. to clarify whether they refer to genes or proteins.
- In Figure 4E, what does the immunoblot in the last line represent?
- We apologize for the missing information in Figure 4E. In response to the reviewers’ comment, we have added α-Actinin. The figure has been revised.
- In Figure 7I, it is not clear how much the scale bar represents the display.
- We thank you for the reviewer’s kind comment. The scale bar represents 30 μm. As suggested, the legend for the Figure 7 has been revised.
- Please add a scale bar to the light microscopy and fluorescence microscopy in Figure 3.
- As suggested, the scale bars were added in Figure 3 and 6. Accordingly, legends have been changed.
- Abbreviations should be defined when they appear in the first time.
- In response to the reviewer’s comment, we have inserted the full name of STAT3 in simple summary section. In addition, we have inserted the full name of TCGA in the Abstract section.
- Please unify the format of references in the article, including the author's name, the case of words in the title of the article, the writing of the name of the journal, and the page number.
- Thank you so much for the reviewer’s detailed comment. All references were standardized using the EndNote format from the instructions for authors of cancers journal website. Nevertheless, we have found a lot of invalid words in reference section, so we corrected our mistakes in reference section.
Round 2
Reviewer 3 Report
Comments and Suggestions for Authors
The authors have addressed all my comments. I recommend accepting it in current form.